# Epstein–Barr virus-based plasmid enables inheritable transgene expression in mouse cerebral cortex

Tomoko Satake[ID]¤*

Molecular Cellular Biology Laboratory, Graduate School of Medical Life Science, Yokohama City University, Yokohama, Japan

¤ Current address: Homeostatic Mechanism Research Unit, Institute of Innovative Research, Tokyo Institute of Technology, Yokohama, Japan
* satake.t.aa@m.titech.ac.jp

## Abstract

Continuous development of the cerebral cortex from the prenatal to postnatal period depends on neurons and glial cells, both of which are generated from neural progenitor cells (NPCs). Owing to technical limitations regarding the transfer of genes into mouse brain, the mechanisms behind the long-term development of the cerebral cortex have not been well studied. Plasmid transfection into NPCs in embryonic mouse brains by *in utero* electroporation (IUE) is a widely used technique aimed at expressing transgenes in NPCs and their recent progeny neurons. Because the plasmids in NPCs are attenuated with each cell division, the transgene is not expressed in their descendants, including glial cells. The present study shows that an Epstein–Barr virus-based plasmid (EB-oriP plasmid) is helpful for studying long-term cerebral cortex development. The use of the EB-oriP plasmid for IUE allowed transgene expression even in the descendant progeny cells of adult mouse brains. Combining the EB-oriP plasmid with the shRNA expression cassette allowed examination of the genes of interest in the continuous development of the cerebral cortex. Furthermore, preferential transgene expression was achieved in combination with cell type-specific promoter-driven transgene expression. Meanwhile, introducing the EB-oriP plasmid twice into the same individual embryos during separate embryonic development stages suggested heterogeneity of NPCs. In summary, IUE using the EB-oriP plasmid is a novel option to study the long-term development of the cerebral cortex in mice.

## Introduction

The adult cerebral cortex in mammals is composed of a diverse array of neurons and glial cells. The fundamental structure is formed during the prenatal period by young neurons generated from neural progenitor cells (NPCs), including radial glial cells (RGCs) in the ventricular and subventricular zones of lateral ventricles [1–3]. NPCs not only self-renew by mitosis, but also generate post-mitotic immature neurons by symmetric and asymmetric cell divisions [2]. After the neurogenesis phase, NPCs switch to produce mainly glial cells postnatally [4–6].

**Data Availability Statement:** All relevant data are within the manuscript and its Supporting Information files.

**Funding:** This research was supported by Yokohama Academic Foundation (https://

yokogaku.com/), the Ichiro Kanehara Foundation for the Promotion of Medical Sciences and Medical Care (https://www.kanehara-zaidan.or.jp/index.html), and Takeda Science Foundation (https://www.takeda-sci.or.jp/). The funders had no role in study design, data collection and analysis, decision to publish, or preparation of the manuscript.

**Competing interests:** The authors have declared that no competing interests exist.

Thus, NPCs are thought to alter their properties to generate different types of progeny cells throughout development [7, 8]. Moreover, NPCs are heterogeneous in terms of their differentiation and proliferation potential [7, 9]. These properties of NPCs result in continuous development of the cerebral cortex from the prenatal to postnatal period. However, the mechanisms behind the long-term development of the cerebral cortex are currently not comprehensively understood.

In the early postnatal cortex, neurons develop elaborate dendrites, and a large number of glial cells (including astrocytes and oligodendrocytes) are generated from NPCs, resulting in a rapid increase in the size of the cerebral cortex [10]. In the ventricular–subventricular zone (V-SVZ), multiciliated ependymal cells called E1 cells are also generated from NPCs [7, 11]. Another important type of postnatal progeny of NPCs is the adult neural stem cells called B1 cells. B1 cells in turn give rise to transient amplifying progenitors known as C cells, which then produce a large number of proliferative immature neurons called A cells [9]. A cells migrate in the rostral migratory stream (RMS) and differentiate into local interneurons of the olfactory bulb (OB) [12]. Impairments in neuron development and gliogenesis lead to alterations in the architecture and functions of the cerebral cortex. These are thought to underlie human neurodevelopmental diseases such as postnatal microcephaly and epilepsy [10, 13, 14]. Although genetic studies of patients have revealed the genes involved in postnatal development, their roles in brain development have not been investigated in detail [14, 15]. This is at least partially due to technical limitations regarding *in vivo* gene transfer into the brains of model mammals.

*In utero* electroporation (IUE) is a technique that is widely used to introduce plasmids into NPCs in the brains of embryonic mice [16–18]. To perform IUE targeting the cerebral cortex, the plasmid solution is injected into the lateral ventricle (LV) of a mouse embryo in the uterus, followed by the application of electric pulses [16, 17, 19]. In general, transfected plasmids neither replicate during the mitotic phase nor are actively retained in the host genome. Therefore, transgene expression from transfected plasmids in NPCs is transient because the plasmids are attenuated with repeated cell division [18]. Neurons generated from recently transfected NPCs, however, harbor plasmids for a long time because they are post-mitotic [18]. Consequently, IUE is suitable for transgene expression in NPCs and their recent progeny neurons. Meanwhile, a genome-integrative plasmid system using the transposon for IUE can enable transgene expression in NPCs and their progeny over a long period of time [20–22]. Transgene integration, however, may impair the integrity of the host genome as the integration position and copy number cannot be controlled [23]. Therefore, a genome-integrative plasmid system is not very suitable for overexpression and RNAi-mediated knockdown and rescue experiments. The development of an inheritable but non-genome-integrating plasmid for IUE might be beneficial to manipulate proliferative cells including NPCs and glial cells.

The EB-oriP plasmid, which encodes Epstein–Barr virus nuclear antigen-1 (EBNA1) and contains the oriP DNA sequence, can persist in host cells as an episome even after successive cell divisions [24, 25]. When the host cell is in a quiescent state, the EB-oriP plasmid is anchored to the host genome through interaction between EBNA1 protein and oriP [26]. During the mitotic phase, the EB-oriP plasmid is replicated using oriP and segregated to daughter cells via association with the host genome [26, 27]. Therefore, the EB-oriP plasmid persists not only in the transfected cells but also in their progeny cells over a long period of time without integration into the host genome. In this study, the EB-oriP plasmid was used for IUE and the transfected mice were analyzed during postnatal development. The results showed that IUE with the EB-oriP plasmid allows transgene expression in all known NPC progeny for at least 2 months after IUE. The EB-oriP plasmid may be applied in conditional transgene expression using transgene expression cassettes with cell type-specific promoters and in knockdown experiments using shRNA expression cassettes. It is also suggested that the heterogeneity of

NPCs regulates their susceptibility to electroporation, which is related to the ability to generate astrocytes in postnatal brains.

Taking the obtained findings together, IUE using the EB-oriP plasmid represents a novel approach with a wide range of applications for studying long-term cerebral development in mice.

## Results

### IUE of the EB-oriP plasmid allows transgene expression in all known NPC progeny in adult mouse brain

First, this study tested whether postnatal NPC progeny express the EB-oriP plasmid transgene introduced by IUE. For this purpose, a green fluorescent protein (GFP) expression cassette was cloned into the EB-oriP plasmid (pCAG-GFP-EB-oriP, Fig 1A) and used for IUE into the LV of ICR mice at E15. It has been reported that almost all neurons in the superficial layer (layers II–III) express the transgene introduced by IUE from a conventional plasmid at E15 [16, 17]. To observe the postnatal progeny of NPCs, the brains were sampled at postnatal day (P)21 (Fig 1A and 1B), when many postnatal progeny of NPCs are thought to have been generated. In the coronal sections of the cerebral cortex, GFP-labeled neurons were observed in the superficial layer (layers II–III) (Fig 1C and 1D, right, respectively) with their axons extended to the contralateral cortex, similar to the GFP-labeled neurons transfected with the control plasmid (pCAG-GFP, Fig 1A) in transfected brains (Fig 1C and 1D, left, respectively). In control brains, 91.4% of the GFP-labeled cells were neurons, while the remaining 8.1% were cells in the V-SVZ (Fig 1D and 1E, left, respectively, and 1F). In pCAG-GFP-EB-oriP-transfected brains, GFP-labeled cells morphologically distinct from neurons were observed throughout the layers and the corpus callosum (Fig 1D, right). These cells were confirmed to be astrocytes and oligodendrocytes using immunostaining for glial fibrillary acidic protein (GFAP) and oligodendrocyte transcription factor-2 (Olig 2), respectively (Fig 1G and 1H). By counting the GFP-labeled cells with astrocyte morphology, approximately 15.1% of GFP-labeled cells were found to be astrocytes. Thus, it can be concluded that the appearance of astrocytes is a clearly quantifiable feature of pCAG-GFP-EB-oriP transfection (Fig 1F). Furthermore, the proportion of GFP-labeled cells in the V-SVZ was significantly greater (17.0%) than that upon control plasmid transfection (8.1%) (Fig 1E and 1F).

The observations shown in Fig 1 suggest that other NPC progeny in the V-SVZ, namely, E1 cells, C cells, and B1 cells, may similarly be GFP-labeled. As predicted, GFP-labeled multiciliated cells considered E1 cells were observed on the ventricular surface (Fig 2A). Furthermore, Mash1-positive C cells (Fig 2B) and GFAP-positive astrocyte-like cells with long basal processes (Fig 2C) were also confirmed to be GFP-labeled. Another class of postnatal NPC progeny is the interneurons in the OB. These cells are produced in the V-SVZ as doublecortin (DCX)-positive immature neurons called A cells, and then migrate in the RMS to the OB. As shown in Fig 2D, very few GFP-labeled cells were observed in the RMS and OB of the control brains. In contrast, several GFP-labeled cells were observed in both the RMS and OB of pCAG-GFP-EB-oriP-transfected brains (Fig 2E). In the OB, GFP-labeled cells morphologically resembling interneurons were observed (Fig 2F, upper). In the RMS, GFP-labeled cells were confirmed to be DCX-positive migrating A cells (Fig 2F, lower). Furthermore, the existence of GFP-labeled astrocytes, A cells in the RMS, and interneurons in the OB was also confirmed at P56 in pCAG-GFP-EB-oriP-transfected brains (S2A–S2C Fig). Taken together, these observations indicate that the transgene expression in all known NPC progeny was achieved by IUE using the EB-oriP plasmid, and the expression level was stable for at least 2 months after IUE.

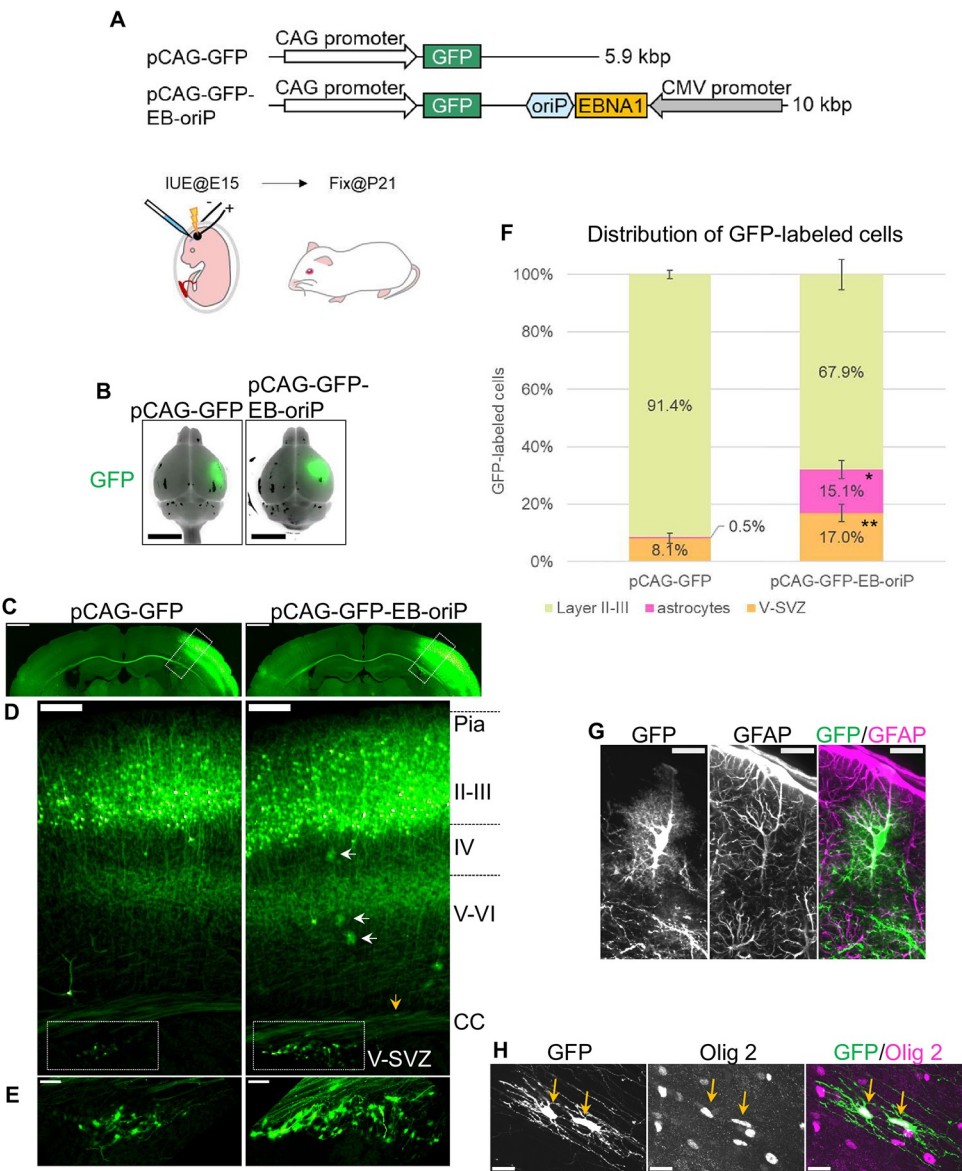

**Fig 1. Transgene expression in not only neurons but also astrocytes and oligodendrocytes by IUE of the EB-oriP plasmid in young adult mice.** (A) Schematic illustration of the plasmid encoding EBNA1 and oriP (pCAG-GFP-EB-oriP), in addition to GFP, and control plasmid (pCAG-GFP). *In utero* electroporation into the lateral ventricle of ICR mice was performed at E15 and the brains were analyzed at P21. (B) Merged images of GFP fluorescence and bright-field image of transfected brains observed at P21. Scale bars: 5 mm. (C) GFP fluorescence images of coronal sections from transfected brains shown in (B). Scale bars: 1 mm. (D) Magnified images of the boxed regions in (C). White arrows and yellow arrow indicate GFP-labeled cells that morphologically appeared to be astrocytes and an oligodendrocyte, respectively. The boxed regions indicate the V-SVZ and are magnified and shown in (E). Pia, pial side; II–III, layers II–III; IV, layer IV; V–VI, layers V–VI; V-SVZ, ventricular–subventricular zone. Scale bars: 200 μm. (E) Magnified images of the boxed regions in (D). Scale bars: 100 μm. (F) Quantification of the distribution of GFP-labeled cells (pCAG-GFP, n = 2114, from four mice; pCAG-GFP-EB-oriP, n = 2601, from five mice). Two to four sections from the central part of the transfected region per animal were counted. Data represent mean ± SEM. *P < 2.2e−16, compared with pCAG-GFP (Fisher's exact test). (G and H) Representative images of GFP-labeled cells in pCAG-GFP-EB-oriP-transfected brain that appeared to be astrocytes immunostained for GFAP (G), and oligodendrocytes immunostained for Olig2 (H, indicated by arrows). Scale bars: 20 μm in (G) and (H).

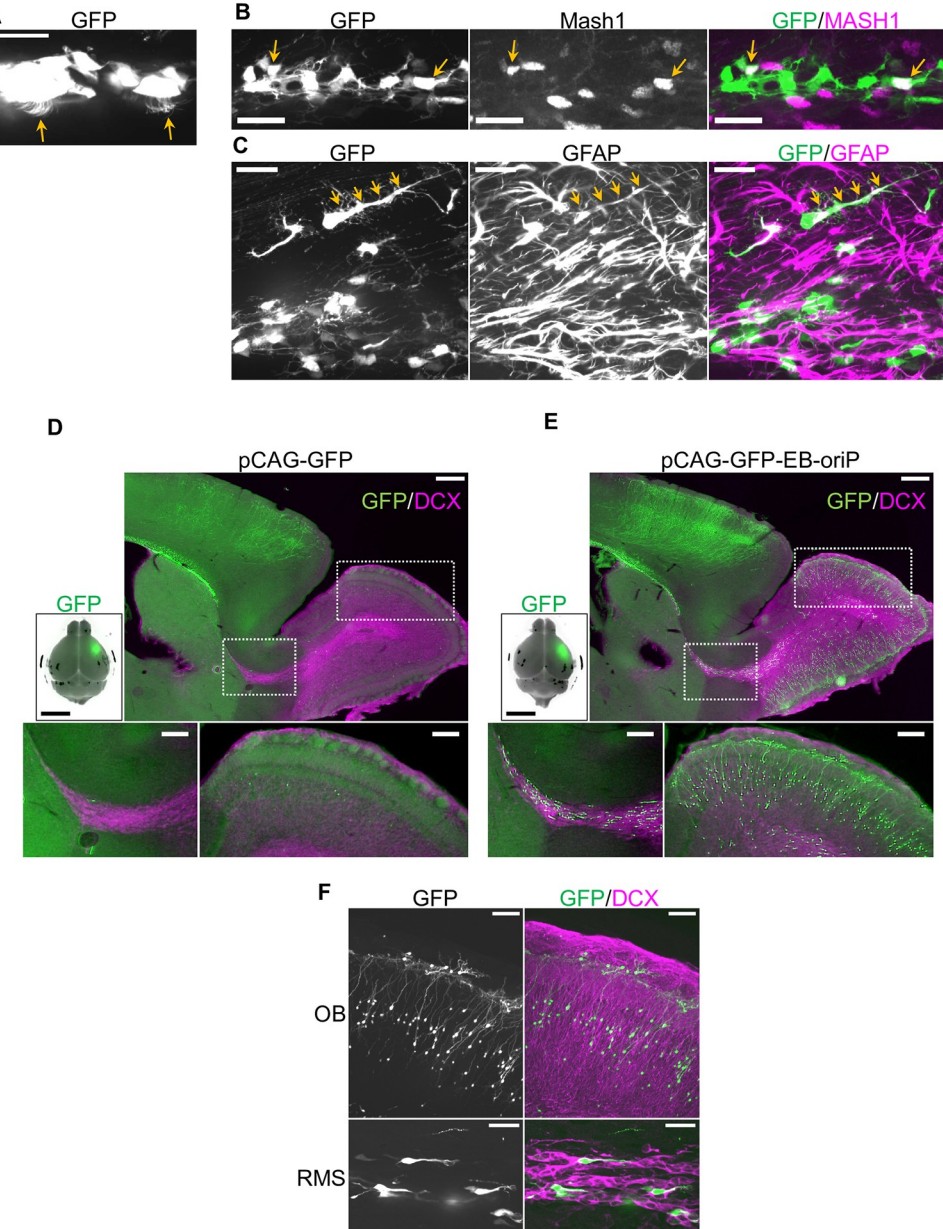

**Fig 2. Transgene expression in E1 cells, C cells, B1 cells, and A cells by IUE of the EB-oriP plasmid in young adult mice.** (A–C) Representative images of GFP-labeled cells in the brains transfected at E15 with pCAG-GFP-EB-oriP and observed at P21. (A) GFP-labeled cells appeared to be E1 cells (indicated by arrows), (B) Mash1-immunopositive C cells (indicated by arrows), and (C) GFAP-immunopositive B1 cell and its long process (indicated by arrows). Scale bars: 25 μm. (D and E) Sagittal sections of the brains transfected at E15 with pCAG-GFP (D) and pCAG-GFP-EB-oriP (E) and fixed at P21 were immunostained for DCX. Merged images of GFP fluorescence and bright-field image of the brains used for sections are shown in upper left panels. The boxed regions in RMS (left) and OB (right), respectively, in upper right panels were magnified and shown in lower panels. Scale bars: 5 mm in upper left panel, 500 μm in upper right panel, 200 μm in lower panels. (F) Representative magnified images of GFP-labeled cells in OB (upper) and RMS (lower) in the section of pCAG-GFP-EB-oriP-transfected brain immunostained for DCX. Scale bars: 100 μm in upper panels and 25 μm in lower panels.

## EBNA1 and the DNA sequences of the EB-oriP plasmid have little effect on the progeny production of NPCs

The EBNA1 expression and nuclear localization were also confirmed in GFP-labeled neurons in pCAG-GFP-EB-oriP-transfected brains at P56, which showed similar morphology to GFP-labeled neurons in control brains (S1D Fig). Thus, 2-month long-term EBNA1 expression appeared to have little effect on the destination, axon projection, and morphological development of layer II–III neurons. Furthermore, given that astrocytes, A cells, C cells, and B1 cells are proliferative cells, the pCAG-GFP-EB-oriP transgene transfected into NPCs appeared to persist and was inherited by their progeny. Indeed, EBNA1 expression in GFP-labeled astrocytes, A cells, interneurons in the OB, and E1 cells was also confirmed at P56 (S1E Fig). However, the possibility that the production of NPC progeny is affected by EBNA1 and/or other DNA sequences in the plasmid cannot be ruled out. To test this hypothesis, a plasmid encoding only GFP and EBNA1 (pCAG-GFP-EB, lacking the oriP sequence), or only GFP and oriP (pCAG-GFP-oriP, lacking EBNA1), was used for IUE, and the transfected brains were observed at P21 and P56 (S2A–S2C Fig). In both pCAG-GFP-EB- and pCAG-GFP-oriP-transfected brains at P21 (S2B Fig) and P56 (S2C Fig), almost all GFP-labeled cells were layer II–III neurons. In contrast, very few GFP-labeled cells were found in the V-SVZ and no GFP-labeled astrocytes were found despite EBNA1 expression (S2D Fig, left), suggesting that EBNA1 could have been expressed in NPCs. Furthermore, very few GFP-labeled migrating A cells were observed in the RMS and interneurons in the OB at both P21 and P56 (S2E and S2F Fig). These results support the notion that the EBNA1 and the DNA sequences of the EB-oriP plasmid have little effect on the progeny production of NPCs over at least 2 months.

The inheritability of the EB-oriP plasmid have been thought to be unviable in several kinds of rodent cells [25, 28–30]. Nonetheless, we tested the EB-oriP plasmid with C57BL6J and ICR mice, and obtained similar results (S3 Fig). Our data indicate that the EBNA1-oriP plasmid can be used for the expression of transgenes in NPC progeny in ICR and C57BL6J mice.

## GFP-labeled cells in the V-SVZ include RGCs, which may transform into astrocytes and B1 cells during early postnatal period

Next, pCAG-GFP-EB-oriP-transfected brains were observed during the early postnatal period. At P7, in addition to the superficial neurons, several GFP-labeled cells were observed in the V-SVZ in both control and pCAG-GFP-EB-oriP-transfected brains (Fig 3A, boxed regions). In early postnatal brains, some of the RGCs still have long basal processes attached to the pial surface by the endfeet [31]. Consistent with this, nestin-positive basal processes and endfeet indeed showed GFP fluorescence at the pial side in pCAG-GFP-EB-oriP-transfected brains (Fig 3B, right), whereas very few GFP-labeled basal processes and endfeet were observed in control brains (Fig 3B, left). Simultaneously, GFP-labeled astrocytes were commonly observed both in the cortex and at the pial surface in pCAG-GFP-EB-oriP-transfected brains (Fig 3A, right), some of which were GFAP-positive (Fig 3C). At P14, GFP fluorescence vanished from basal processes, while numerous GFP-labeled astrocytes were observed in pCAG-GFP-EB-oriP-transfected brains (Fig 3D, right). In V-SVZ, GFP-labeled cells were still observable, although their numbers were reduced (Fig 3D, right, boxed region). These observations are consistent with and support the current model that RGCs transform into astrocytes and B1 cells during the early postnatal period [7].

## NPC populations susceptible to electroporation at E13 and E15 are heterogeneous and have different progeny-producing properties

Embryonic NPCs in the V-SVZ have been thought to generate neurons destined for lower and upper layers and glial cells, serially and continuously, during cortical development. Therefore,

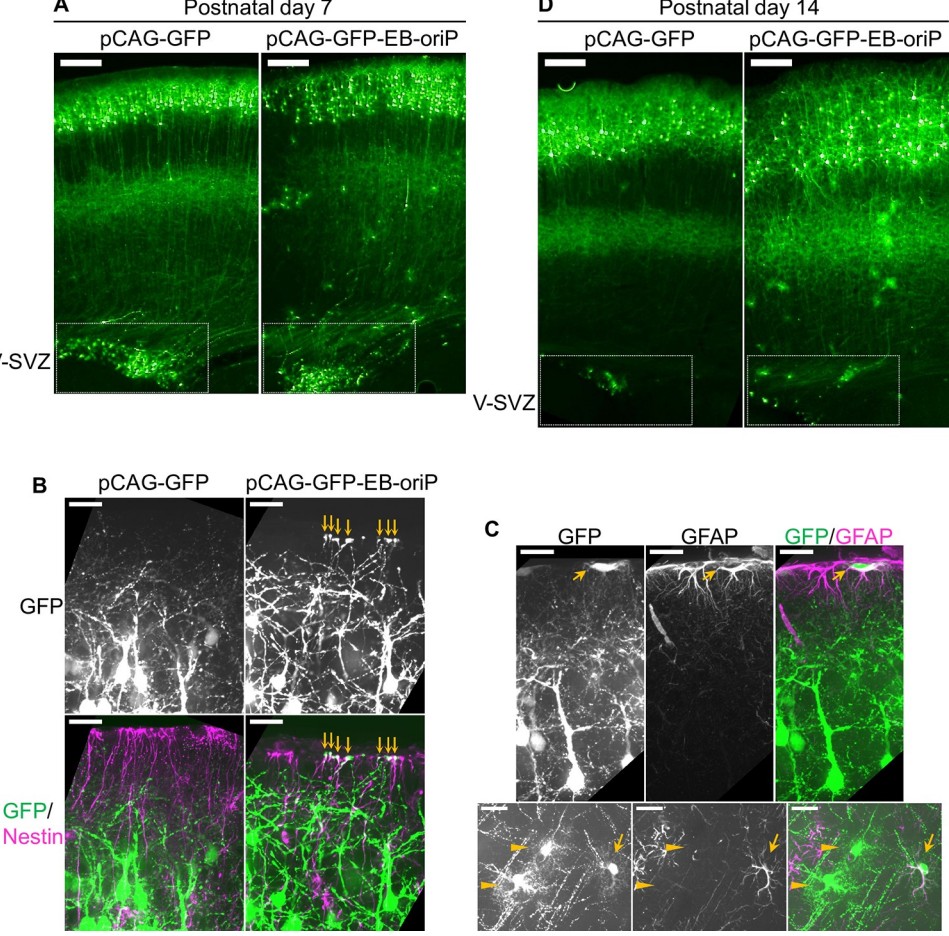

**Fig 3. Transgene expression in radial glial cells by IUE of the EB-oriP plasmid in early postnatal mice.** (A and D) GFP fluorescence images of coronal sections at P7 (A) and P14 (D) from the brains transfected at E15 with pCAG-GFP and pCAG-GFP-EB-oriP. Scale bars: 200 μm. (B) GFP-labeled cells in the superficial region at P7 and immunostaining for Nestin. Arrows indicate the parts of GFP-labeled cells that are Nestin-positive, which appeared to be radial glial endfeet. Scale bars: 25 μm. (C) GFP-labeled cells at P7 morphologically appeared to be astrocytes immunostained for GFAP. Arrows in upper and lower panels and arrowheads in lower panels indicate GFAP-positive or -negative cells, respectively. Scale bars: 25 μm.

IUE of the EB-oriP plasmid at an earlier embryonic stage is expected to allow transgene expression in neurons across wide areas and in glial cells. To confirm this expectation, IUE was performed at E13 and brains were sampled at P21. First, neurons in the deeper layer (layer IV) were confirmed to be successfully transfected by IUE using the control plasmid (71.5%, Fig 4A, lower panels, and C), instead of GFP labeling of the superficial neurons (5.6%, Fig 4A, bottom right) and the cells in the V-SVZ (0%, Fig 4A, bottom left, boxed region). In pCAG-GFP-EB-oriP-transfected brains, both deeper and superficial neurons, astrocytes, and cells in the V-SVZ were expected to be labeled with GFP. Indeed, in contrast to the findings upon control transfection, GFP-labeled neurons were observed in multiple layers (Fig 4B, lower panels), although there were fewer GFP-labeled neurons in layers II–III than in layer IV (Fig 4B, bottom right). Unexpectedly, GFP-labeled astrocytes (0.9%) and cells in the V-SVZ (2.1%) were rare (Fig 4B, bottom left, boxed region, and 4D). In contrast, in the RMS and OB, GFP-labeled A cells and interneurons were observed (Fig 4F), which matched the results of

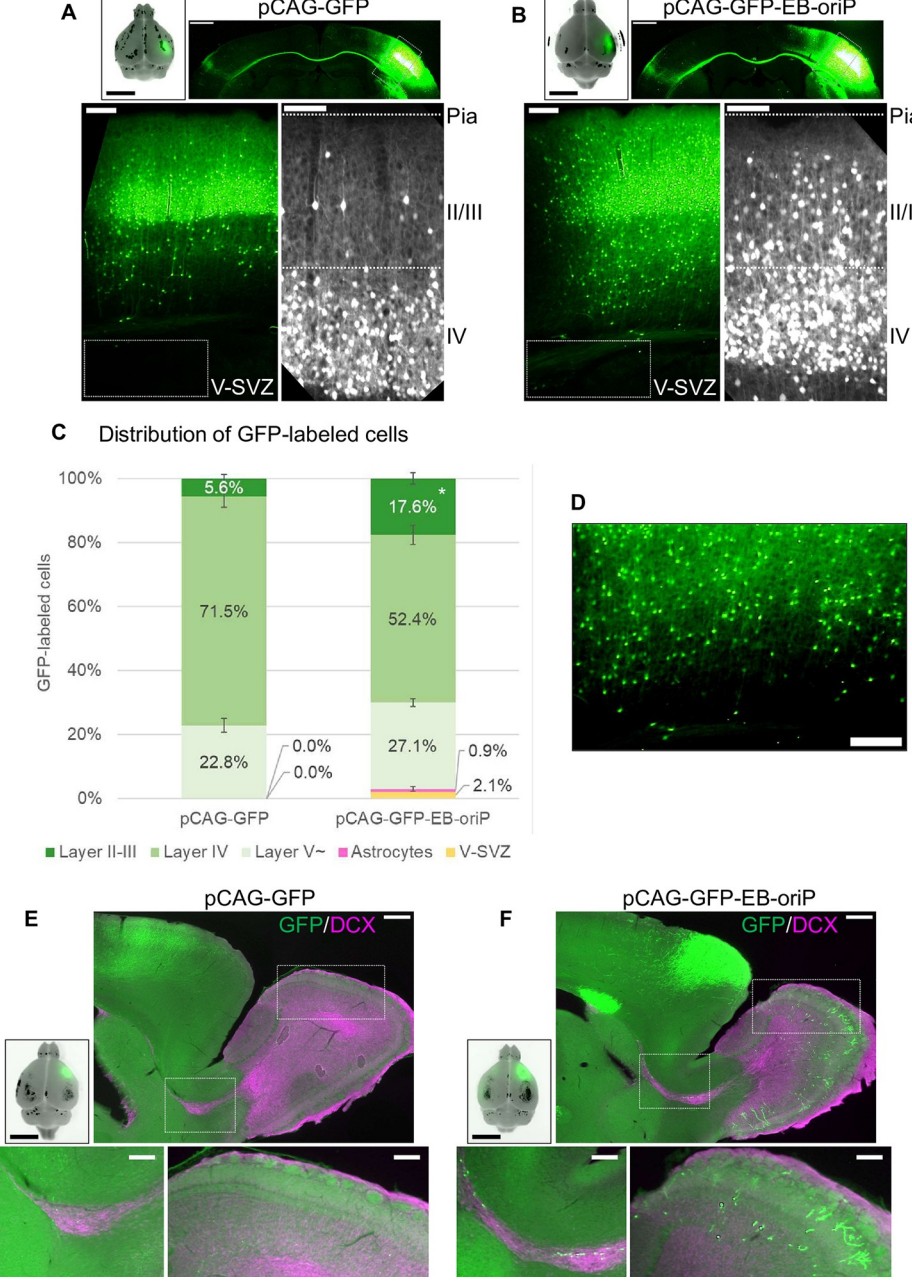

**Fig 4. Transgene expression in neurons in a broad range of layers but few in astrocytes and cells in V-SVZ by IUE of the EB-oriP plasmid at E13.** (A and B) GFP fluorescence images of coronal sections from the brains transfected at E13 with pCAG-GFP (A, upper right) and pCAG-GFP-EB-oriP (B, upper right) and observed at P21. Merged images of GFP fluorescence and bright-field image of the brains used for sections are shown in the upper left panels. The boxed regions were magnified and shown in the lower left panels, respectively. Magnified GFP fluorescence images of the superficial region are shown in the lower right panels. Scale bars: 5 mm in upper left panels, 1 mm in upper right panels, 200 μm in lower left panels, 100 μm in lower right panels. (C) Quantification of the distribution of GFP-labeled cells (pCAG-GFP, n = 2054; pCAG-GFP-EB-oriP, n = 4359, from three mice). Two to three sections from the central part of the transfected region per animal were counted. Data represent mean ± SEM. *P < 2.2e−16, compared with pCAG-GFP (Fisher's exact test). (D) Magnified GFP fluorescence images of the deeper layer region of pCAG-GFP-EB-oriP-transfected brain, showing few GFP-labeled astrocytes. Scale bar: 200 μm. (E and F) Sagittal sections from the brains transfected at E13 with pCAG-GFP (E) and pCAG-GFP-EB-oriP (F) and fixed at P21 were immunostained for DCX. Merged images of GFP fluorescence and bright-field image of the brains used for sections are shown in upper left panels. The boxed regions in RMS (left) and OB (right) were magnified and are shown in lower panels. Scale bars: 5 mm in upper left panel, 500 μm in upper right panel, 200 μm in lower panels.

IUE at E15 (Figs 2E, and 4E). These results suggest that NPC populations susceptible to electroporation at E13 and E15 are distinct from each other, and have different properties regarding the generation of layer II–III neurons in the prenatal period and astrocytes in the postnatal period.

To test this hypothesis, IUE was performed separately at E13 and E15 at the same region of the same individual mouse embryos and the transfected brains were observed at P21 (Fig 5A). To distinguish the progeny cells derived from the NPCs transfected at the different stages, EB-oriP plasmids encoding GFP or red fluorescent protein (RFP) were used at E13 and E15, respectively (Fig 5A). Successful transfection of the plasmids into the same area by separate transfection events was confirmed by fluorescence stereomicroscopy (Fig 5B, left) by comparison with the fluorescence of brains co-transfected at E13 (Fig 5B, right). Indeed, GFP-labeled cells fell within the area occupied by RFP-labeled cells in separately transfected brains (Fig 5C and 5D, left, respectively), whereas GFP- and RFP-labeled cells completely overlapped in co-transfected brains (Fig 5C and 5D, right, respectively). In co-transfected brains, co-labeled cells were often seen in a broad range of layers (Fig 5E, right), whereas co-labeled cells were rare in separately transfected brains (Fig 5E, left). To characterize the population of progeny cells, the distributions of GFP- and RFP-labeled cells were quantified per animal (Fig 5G). The numbers of GFP- and RFP-labeled cells in each layer were counted in the region where GFP-labeled cells were found throughout the layers. In co-transfected brains, the distribution patterns of GFP- and RFP-labeled cells were similar (Fig 5G, right), which is consistent with the results from transfection using pCAG-GFP-EB-oriP alone (Fig 4). In layers II–III, 32.2% and 44.4% of GFP-labeled cells in Mouse 4 and Mouse 5, respectively, were also labeled with RFP, even though the proportion of GFP-labeled cells in this layer was limited (13.8% and 19.2% of total GFP-labeled cells in Mouse 4 and Mouse 5, respectively). In separately transfected brains, GFP-labeled cells showed a similar distribution pattern to co-transfected brains, while the RFP-labeled cells were distributed mainly in layers II–III (85.9% in Mouse 1, 88.3% in Mouse 2 and 84.7% in Mouse 3, Fig 5G, left), consistent with the results from transfection using pCAG-GFP-EB-oriP (Fig 1). Unlike in co-transfected brains, even in the region fully occupied by the RFP-labeled cells (Fig 5E, left), the co-labeled cells only constituted 6.3% in Mouse 1, 2.3% in Mouse 2 and 3.5% in Mouse 3 of GFP-labeled cells in layers II–III (Fig 5G, left) in separately transfected brains. The RFP-labeled astrocytes accounted for 13.9% in Mouse 1, 11.4% in Mouse 2 and 15.3% in Mouse 3, whereas the GFP-labeled astrocytes were only 0.8% in Mouse 1, 0% in Mouse 2 and Mouse 3 (Fig 5G, left), consistent with the results shown in Figs 1 and 4, respectively. These results suggest that NPCs susceptible to electroporation at E13 have less potential to produce upper-layer neurons and astrocytes than NPCs susceptible to electroporation at E15. This is consistent with the patterns observed for GFP- and RFP-labeled cells in V-SVZ (Fig 5F). Because GFP-labeled cells were few in number and distributed within a confined area, the brain sections containing both GFP- and RFP-labeled cells in V-SVZ were limited. As such, fluorescent cells in the V-SVZ were not quantified. Instead, all sections containing both GFP- and RFP-labeled cells in the V-SVZ were observed at high magnification to assess whether they were co-labeled (Fig 5F). In co-transfected brains, co-labeled cells were abundant (Fig 5F, right), even though the numbers of fluorescently labeled cells were small. In contrast, co-labeled cells were rarely found in separately transfected brains, even though GFP- and RFP-labeled cells were readily observable. Indeed, only 15 of 193 GFP-labeled cells in three mice were found to be co-labeled in separately transfected brains, whereas 38 of 58 GFP-labeled cells in two mice were co-labeled in co-transfected brains. These results imply that the NPCs susceptible to electroporation at E13 and/or their progeny may not be susceptible at E15.

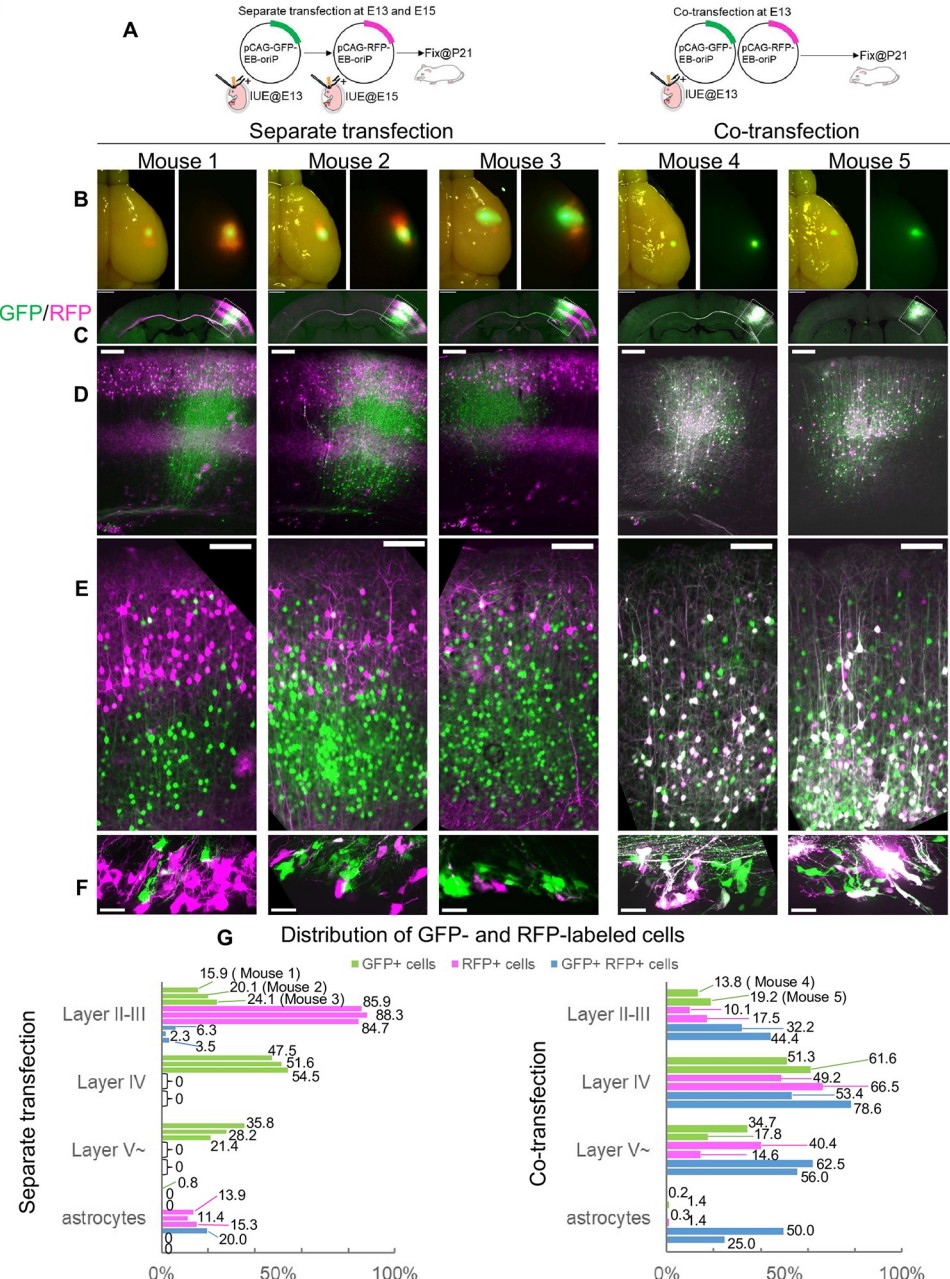

**Fig 5. NPCs susceptible to IUE at E13 might be less susceptible at E15.** (A) Schematic illustration of the experiment. IUE of the EB-oriP plasmids encoding either GFP or RFP was performed twice at E13 and E15 (separate transfection, left) or once at E13 (co-transfection, right). Transfected brains were analyzed at P21. (B) Individual images of the successful transfection. Merged fluorescence image of GFP and RFP (right panels) were merged with bright-field image (left panels). (C–F) (C) Merged fluorescence images of GFP and RFP of the coronal sections from the brains with separate transfection (left) and co-transfection (right). The boxed regions were magnified and are shown in (D). Magnified images of the superficial regions (E) and V-SVZ (F). Scale bars: 1 mm in (C), 200 μm in (D), 100 μm in (E), 25 μm in (F). (G) Quantification of the distributions of GFP- and RFP-labeled cells per animal (separate transfection, Mouse 1, n = 604 (GFP), n = 710 (RFP); Mouse 2, n = 2369 (GFP), n = 446 (RFP); Mouse 3, n = 2032 (GFP), n = 333 (RFP); co-transfection, Mouse 4, n = 854 (GFP), n = 606 (RFP); Mouse 5, n = 281 (GFP), n = 212 (RFP)). Two to five sections from the central part of the transfected region per animal were counted.

## The EB-oriP plasmid is a suitable tool for cell type-selective transgene expression in mouse cerebral cortex

Cell type-selective transgene expression mediated by the EB-oriP plasmid may be used in conjunction with cell type-specific promoters. To demonstrate this, two EB-oriP-based plasmids named Acceptor and Donor were constructed. The Acceptor plasmid encodes GFP and RFP (Fig 6A), where GFP expression requires the excision of a loxP-flanked STOP sequence by Cre recombinase (Cre), and the RFP expression is driven by the upstream activation sequences (5xUAS). The Donor plasmid encodes Cre and a fusion protein consisting of the Gal4 DNA-binding domain and the VP16 transcriptional activation domain (GAL4-VP16), which binds 5xUAS to drive transcription. The Cre expression is driven by cytomegalovirus (CMV) promoter, while the GAL4-VP16 expression is driven by mouse *GFAP* (*mGFAP*) promoter. Thus, the GFP expression is expected to occur only in the cells co-transfected with both plasmids. Furthermore, GFP-labeled cells are also labeled with RFP when the *mGFAP* promoter is active, specifically RGCs in early postnatal brains, and some astrocytes and B1 cells in adult brains. These plasmids were used for IUE at E15, and the transfected brains were observed at P5, P14, and P21 (Fig 6B). During all stages, layer II–III neurons, astrocytes, and the cells in the V-SVZ were found to be GFP-labeled (Fig 6B). As expected, the process and cell body of RGCs in the V-SVZ were clearly co-labeled with RFP at P5 (Fig 6C, upper and lower), while no GFP-labeled neurons were co-labeled with RFP (Fig 6C, upper). Additionally, the majority of GFP-labeled astrocytes in the cortex were co-labeled with RFP during all stages (Fig 6B and 6D, upper). In the V-SVZ of P21 brain, co-labeled cells with bushy processes resembling B1 cells were found (Fig 6D). To investigate whether RFP labeling corresponds to GFAP protein expression, RFP-labeled cells from P14 brains were immunostained for GFAP. In the cortex, both GFAP-positive (Fig 6E, upper) and GFAP-negative cells (Fig 6E, lower) were found among the RFP-labeled cells. In contrast, in the V-SVZ, all RFP-labeled cells were GFAP-positive (Fig 6F). This suggests that the *mGFAP* promoter activity in the plasmid is not fully identical to the endogenous GFAP protein expression in astrocytes, whereas it almost completely corresponds to that in the B1 cells. These results indicate that co-transfection of the Acceptor and Donor plasmids by IUE has the potential to be used for cell type-selective transgene expression in mouse cerebral cortex.

## The EB-oriP plasmid may be used as a tool to continuously examine gene functions during development of the cerebral cortex

Finally, the feasibility of using the EB-oriP plasmid as a tool to examine gene functions in cerebral cortex development was tested. For this purpose, a gene for which mutations in humans are associated with abnormal postnatal brain development, but with no clarified role in either brain development or pathogenesis, was chosen as a candidate, namely, tubulin-cofactor D (TBCD). Mutations in TBCD cause early postnatal-onset impaired brain growth and development accompanied by various neurological symptoms [32–36]. To examine the role of TBCD, two shRNAs (shTBCD#1 and shTBCD#2) against it were designed and incorporated into the pCAG-GFP-EB-oriP plasmid, and then used for IUE at E15 (Fig 7A). Immunostaining of transfected brains was not performed because no antibodies for TBCD suitable for immunostaining were obtained. Instead, the knockdown efficiency of the shRNAs was tested by assaying protein levels of co-transfected V5-tagged mouse TBCD in HEK293T cells (Fig 7B). When observed at P14, GFP fluorescence was prominent in non-silencing shRNA (NS)-transfected brains, while no GFP fluorescence was detected in shTBCD #1- and shTBCD#2-transfected brains (Fig 7C, upper right panels). Notably, GFP fluorescence of the same mice at P1 was prominent when observed from the skull and skin (Fig 7C, upper left panels), suggesting that

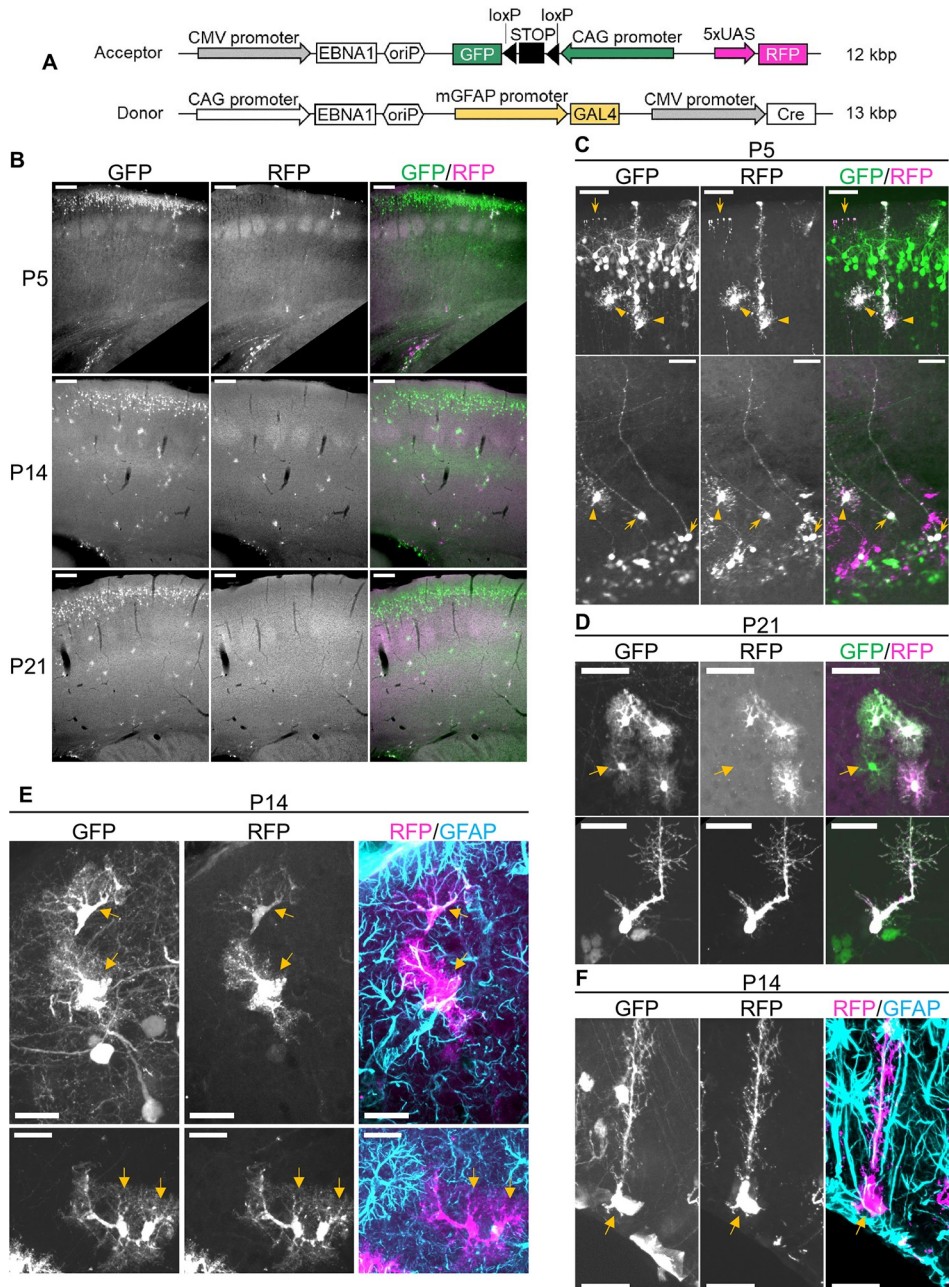

**Fig 6. Conditional transgene expression by the EB-oriP plasmid is possible in combination with cell type-selective promoter.** (A) Schematic illustration of the constructed plasmids named Acceptor and Donor. These were co-transfected by IUE at E15 and the brains were analyzed at P5, P14, and P21. Acceptor encodes GFP and RFP. The GFP expression driven by the ubiquitous CAG promoter requires excision of the loxP-flanked STOP sequence by Cre, and the RFP expression is driven by 5xUAS. Donor encodes Cre and GAL4-VP16, which activate transcription from 5xUAS. The Cre expression is driven by CMV promoter, while the GAL4-VP16 expression is driven by mGFAP promoter. Thus, GFP is expressed only in the cells co-transfected with both plasmids. Furthermore, RFP is expressed in the GFP-labeled cells in which the mGFAP promoter is active. (B) Fluorescence images of GFP and RFP of coronal sections from transfected brains at the indicated age. Scale bars: 200 μm. (C) Representative images of GFP- and RFP-labeled cells at P5. Doubly labeled cells appeared to be astrocytes in the superficial region (C, upper) and V-SVZ (C, lower) is indicated by arrowheads. Doubly labeled cell appeared to be radial glia (C, lower) and its endfeet (C, upper) are indicated by arrows. Scale bars: 50 μm. (D) Representative images of GFP- and RFP-labeled cells at P21. Either doubly labeled or only GFP-labeled astrocytes (indicated by arrow) in cortex (upper). Doubly labeled cell appeared to be a B1 cell in V-SVZ (lower). (E and F) GFP- and RFP-labeled cells at P14 and immunostaining for GFAP. Doubly

labeled astrocytes are indicated by arrows, which were GFAP-positive (E, upper) and -negative (E, lower). Doubly labeled cell appeared to be a B1 cell in V-SVZ, which was GFAP-positive (F). Scale bars: 25 μm.

faint GFP fluorescence is not the result of low transfection efficiency. Therefore, the GFP fluorescence appears to have diminished by 14 days after birth in shTBCD-transfected brains. As expected, in NS plasmid-transfected brains, neurons in the superficial layer, astrocytes, and the cells in the V-SVZ were found to be GFP-labeled (Fig 7C, left). There was a dense population of GFP-labeled neurons in the superficial layer. A similar dense population of GFP-labeled cells was observed in the V-SVZ. The GFP-labeled astrocytes were scattered throughout the layers and near the V-SVZ. In shTBCD-transfected brains (Fig 7C, center and right), GFP-labeled cells were observed, but their total numbers and density were markedly reduced, which is consistent with the faint GFP fluorescence seen in transfected brains. Particularly in shTBCD#1-transfected brains, GFP-labeled cells morphologically resembling neurons were often observed in the deeper layer, whereas GFP-labeled astrocytes were rare (Fig 7C, center). The effect of shTBCD#2 transfection was milder than that of shTBCD#1 in terms of the number and density of GFP-labeled cells (Fig 7C, right). Nevertheless, the GFP fluorescence of superficial neurons was visibly weaker than that of other GFP-labeled cells, including astrocytes and cells in the V-SVZ (Fig 7C, right). To confirm the effect of shTBCD transfection in detail, the distributions of GFP-labeled cells were quantified and compared to the findings upon NS transfection. The results showed a clear trend of a lower proportion of neurons in the superficial layers (layers II–III, 58.9% in NS, 57.3% in shTBCD#1, 43.9% in shTBCD#2) and a significantly lower proportion of astrocytes (20.8% in NS, 11.9% in shTBCD#1, 18.4% in shTBCD#2). Mislocalized neurons appeared in shTBCD#1-transfected brains (5.8%) but were rare in shTBCD#2-transfected ones (0.3%). Consequently, the proportions of cells in the V-SVZ were significantly higher (20.2% in NS, 37.5% in shTBCD#2). Next, it was investigated whether these trends are also present earlier in development at P7. shTBCD#1 was not used here because it was assumed to induce rapid reduction of GFP-labeled cells during the postnatal period, which is not suitable for quantification and subsequent analyses. As in the observation at P14, NS-transfected brains showed prominent GFP fluorescence at P7 (Fig 7D, left, upper). GFP fluorescence of shTBCD#2-transfected brains was barely visible at P7 (Fig 7D, right, upper right), although it was comparable to that of NS-transfected brains at P1 (Fig 7D, right, upper left). Notably, as at P14, GFP fluorescence of superficial neurons was visibly weaker, coinciding with the faint GFP fluorescence of the brain. The proportion of GFP-labeled astrocytes was significantly lower (25.0% in NS, 13.6% in shTBCD#2), which consequently slightly raised the proportion of GFP-labeled superficial neurons (39.7% in NS, 45.0% in shTBCD#2) and cells in the V-SVZ (35.1% in NS, 37.4% in shTBCD#2). These data suggest that the effect of shTBCD#2 transfection appears by P7 and progresses with age. Thus, these results suggest that TBCD expression in neurons may be essential for neural migration and survival during the early postnatal period. Furthermore, considering that a lower proportion of GFP-labeled astrocytes was observed in two independent shRNA-transfected samples, TBCD may also be essential for astrocyte differentiation, survival, and proliferation.

Taking these findings together, the EB-oriP plasmid-mediated shRNA transgenic technology may become a tool to continuously examine gene functions in development of the cerebral cortex.

## Discussion

In this study, the EB-oriP plasmid introduced by IUE was demonstrated to allow persistent and inheritable transgene expression in all known NPC progeny in the cerebral cortex of mice.

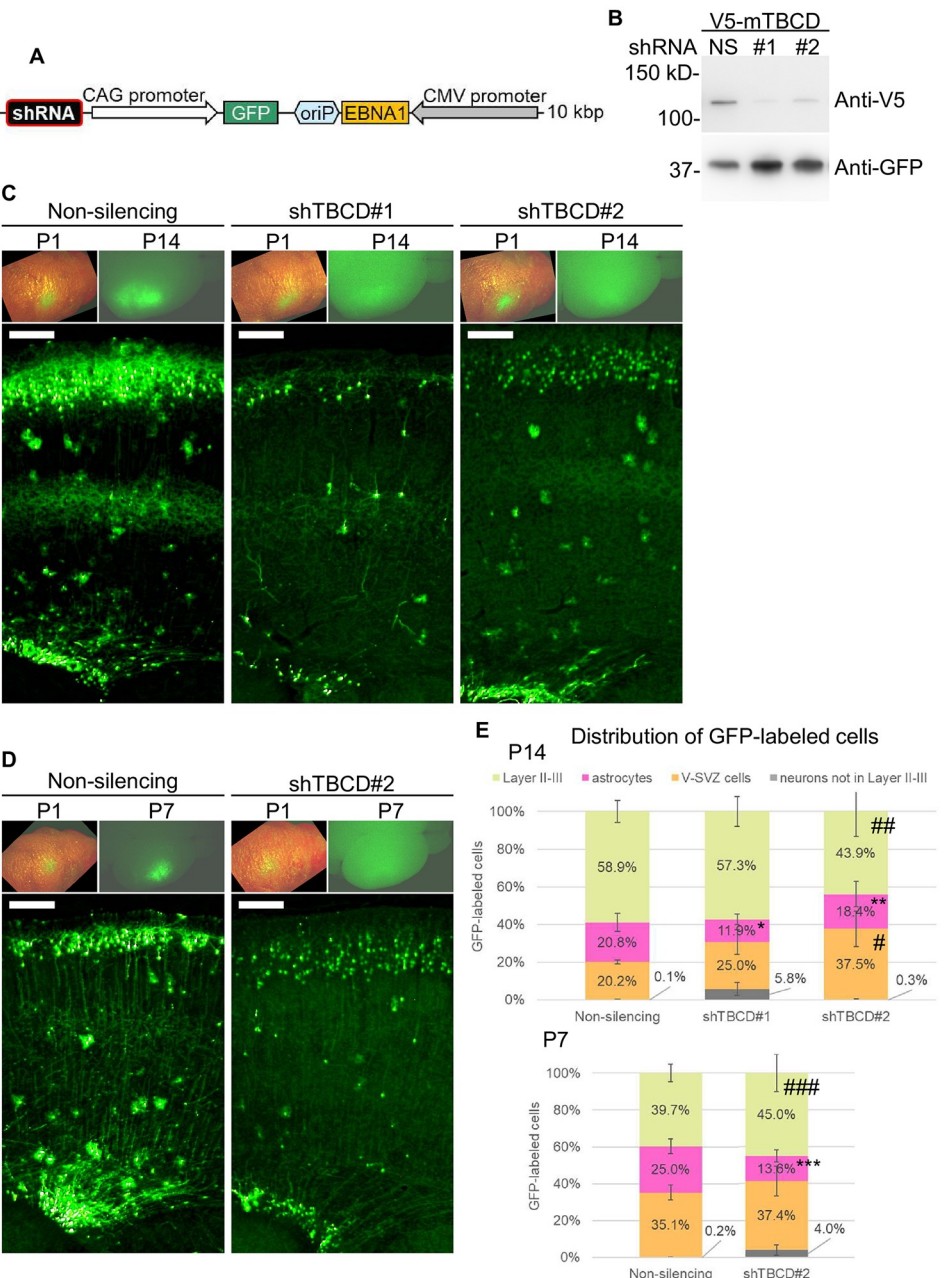

**Fig 7. IUE of the EB-oriP plasmid encoding shRNA against TBCD suggests the role of TBCD in postnatal cerebral cortex.** (A) Schematic illustration of the EB-oriP plasmid encoding shRNA expression in addition to GFP expression. The expression cassette of shRNA targeting mouse TBCD or non-silencing control was cloned into pCAG-GFP-EB-oriP plasmid (Fig 1A). IUE of these plasmids was performed at E15 and the brains were analyzed at P7 and P14. (B) Lysates prepared from HEK293T cells co-transfected with plasmids for V5-tagged mouse TBCD and plasmid shown in (A) (NS, non-silencing; #1 and #2, two independent shRNAs against TBCD) were analyzed by western blot with the indicated antibodies. (C and D) Representative GFP fluorescence of the head of a mouse transfected with the indicated plasmid at P1 was merged with the bright-field image (upper left). GFP fluorescence of the brain shown in left panel at P14 (C, upper right) and P7 (D, upper right). Representative GFP fluorescence of coronal section from the brain is shown in upper right panel. (E) Quantification of the distribution of GFP-labeled cells at P14 (upper) and P7 (lower) (P14: NS, n = 4085, from four mice; shTBCD#1, n = 926, from three mice; shTBCD#2, n = 1873, from four mice, P7: NS, n = 2768, from four mice; shTBCD#2, n = 1171, from three mice). Two to three sections from the central part of the transfected region per animal were counted. Data represent mean ± SEM. $^*$P = 9.642e-09, $^{**}$P = 0.0003689, $^{***}$P = 8.948e-15, #P = 1.22e-12, ##P = 0.0119, ###P = 0.003212, compared with NS (Fisher's exact test).

It was also shown that cell type-preferential transgene expression in some of the astrocytes could be achieved by the EB-oriP plasmid in combination with the mGFAP promoter. Furthermore, the shRNA expression cassette introduced by the EB-oriP plasmid was shown to be a potentially viable tool to continuously determine gene functions during development of the cerebral cortex. Moreover, the experimental data in this study suggested that NPCs contain sub-populations with different levels of susceptibility to electroporation, which is related to differences among the progeny cell types. In summary, the introduction of the EB-oriP plasmid using IUE represents a novel transgenic tool with a wide range of applications for studying cerebral development in mice.

## EB-oriP plasmid introduced by IUE enables persistent and inheritable transgene expression in all known NPC progeny

IUE has several advantages over virus-based transfection methods, including the simplicity of plasmid preparation and the lack of a need for facilities for infected animals. However, the expression of transgenes introduced by IUE is mostly limited to recently transfected NPCs and their progeny because the plasmids are not maintained during multiple cell divisions. To achieve persistent and inheritable transgene expression, integration of the transgene into the host cell genome is often performed using a virus vector or transposon-mediated plasmid transfection with IUE [20–22, 37], although both methods may disrupt the integrity of the host cell genome. Ideally, persistent and inheritable transgene expression should be achieved without integration into the host cell genome. The EB-oriP plasmids are maintained as episomes and are inheritable in human host cells [24]. It was previously assumed that these properties could not be exhibited in several different rodent cells [25, 28–30]. Nonetheless, in this study, introduction of the EB-oriP plasmid by IUE was shown to be sufficient to mediate transgene expression in all known NPC progeny in mice. This effect persisted for at least 2 months and 3 weeks after IUE for ICR and C57BL6J mice, respectively. Meanwhile, the cell types and mouse strains may also influence the behavior of EB-oriP plasmids in mouse cells. Therefore, it is unclear whether EB-oriP plasmids introduced by IUE will produce similar results in other mouse strains and rodents, or in other mouse tissue progenitor cells.

Although the presence of the EB-oriP plasmid in GFP-labeled cells has not been directly confirmed, the results strongly suggest that EB-oriP plasmids are maintained and inherited in NPCs and their progeny cells. Meanwhile, plasmids lacking either oriP or EBNA1 failed to produce GFP fluorescence in NPC progeny cells (S2 Fig). Therefore, neither the oriP nor the EBNA1 gene appears to strongly affect the production of NPC progeny. However, certain cellular functions are likely to be affected by the EB-oriP plasmid. In fact, the EBNA1 protein has been shown to affect several host proteins and signaling pathways in host cells [38, 39]. Further investigations are required to evaluate the extent of host cell perturbation induced by the EB-oriP plasmid.

Many studies aimed at developing more efficient episomal plasmids have been carried out to improve gene therapy treatments [40]. It has been reported that plasmids with scaffold/matrix attached region (S/MAR) elements are retained in host cells as episomes, and thereby replicate and are inheritable [41–43], although the efficiency of establishment is low [42]. It was also examined whether NPC progeny produce GFP fluorescence using plasmids with an S/MAR-based sequence in mice for IUE at E15 (S4 Fig). Human β-interferon S/MAR element was inserted downstream of the GFP coding sequence lacking a transcription termination signal (pCAG-GFP-S/MAR) to promote S/MAR-mediated plasmid maintenance in the host cell [42]. Using this construct, all superficial neurons became GFP-labeled, but few other progeny cell types, including astrocytes and interneurons in the OB, did so. This suggests that the

efficiency of establishment of human β-interferon S/MAR-based plasmids is extremely low under this set of conditions. It is possible that other S/MAR elements work efficiently in episome maintenance.

## The EB-oriP plasmid may be a novel solution to study cerebral cortex development from the prenatal to postnatal period in mice

NPCs undergo repetitive cell division for self-renewal and neuron generation during the prenatal period, and therefore defects in NPC division and neuron migration are common causes of congenital cortical malformation [14, 15]. Indeed, many genes have been shown to be associated with these processes [14, 15]. Notably, IUE has been employed to identify the functions of the genes causative of defects in prenatal cortical development [44–46]. During the early postnatal period, NPCs generate glial cells to mediate functional and cytoarchitectural development of the cerebral cortex with a rapid increase of cerebral cortex size. Owing to technical limitations regarding the transfer of genes into postnatal brains, the pathogenesis of postnatal-onset genetic disorders of cerebral development has been largely unexplained, even though many causative genes have been reported. Against this background, the role of TBCD was examined using the EB-oriP plasmid with shRNA against TBCD. Biallelic mutations in the TBCD gene cause early postnatal-onset progressive neurodegenerative encephalopathy with postnatal microcephaly in human patients [32–36]. The results of this study suggest that TBCD is involved in astrocyte generation during early postnatal development in mice (Fig 7). Furthermore, TBCD has been suggested to regulate neuronal functions. Taking these findings together, mutations in TBCD may lead to impairments in astrocyte generation and neuron maintenance, thus causing early postnatal-onset microcephaly of the cerebral cortex. These observations may help not only to understand the role of TBCD in cerebral cortex development, but also to explain the mechanism of impaired brain development caused by TBCD mutation in humans. Meanwhile, the experimental results in this study are insufficient to draw definitive conclusions on the physiological function of TBCD. Further detailed investigations are necessary to understand the role of TBCD in brain development and the pathogenetic effect of TBCD mutations in human patients.

In summary, the expression of shRNA using the EB-oriP plasmid introduced by IUE is a useful tool for studying gene functions, including those of other causative genes involved in cerebral cortex development from the prenatal to postnatal period. Furthermore, transgene expression in combination with conditional promoters, such as cell type-specific promoters and drug-inducible promoters, can help to more precisely examine gene functions.

IUE of the EB-oriP plasmid can be an additional experimental manipulation tool for clonal analysis and lineage tracing in combination with Cre-dependent multicolor fluorescent reporter mice [47]. The CreER$^{T2}$ expression using the EB-oriP plasmid introduced by IUE can enable multi-color fluorescent labeling in the progeny cells of transfected progenitor cells at the time of tamoxifen administration. Furthermore, along with conditional fluorescent labeling with CreER$^{T2}$, co-transfection with another EB-oriP plasmid coding a Cre-dependent transgene expression system, such as Cre-dependent excision of the STOP sequence shown in this study, can enable shRNA expression and genome modification by CRISPR/Cas9, as well as simple forced expression. In addition, as mentioned above, cell type-selective manipulation can be achieved by using cell type-specific promoters.

## Heterogeneity of NPCs could be reflected in electroporation susceptibility

The proportion of astrocytes from NPCs transfected with the EB-oriP plasmid using IUE at E13 was significantly lower than that at E15 (Fig 4). Furthermore, the cells in the V-SVZ

transfected at E13 appear to have become less susceptible to IUE at E15 (Fig 5). These results support the notion that the NPC populations susceptible to IUE at E13 and E15 have different properties regarding progeny production. Similar results were obtained from studies using transposon-mediated plasmids in mice and rats [37, 48]. In the V-SVZ, RGCs form a single pseudostratified layer and undergo interkinetic nuclear migration according to their cell cycle phases [49, 50]. Therefore, RGCs indeed show heterogeneity based on their distance from the ventricular surface and the size of the area contacting it. These factors may regulate the cells' susceptibility to electroporation. Thus, it will be intriguing to identify such specific factor(s) determining the susceptibility of NPCs to electroporation, which may offer clues to understand the heterogeneity of NPCs.

## Materials and methods

### Animals

All protocols were approved by and performed in accordance with the guidelines established by the Institutional Animal Care and Use Committee (IACUC) at Yokohama City University, Medical Life Science (Approval No. T-A-19-007). The day of plug identification was designated as embryonic day (E)0, and the day of birth was designated as postnatal day (P)0. Male and female mice were used equally in all experiments.

### *In utero* electroporation

For *in utero* electroporation, ICR and C57BL6J mice were purchased from Japan SLC (Hamamatsu, Japan). Two to four pregnant mice were housed together for at least 2 days before surgery. The mice were anesthetized by the intraperitoneal injection of a mixture of three different anesthetic and analgesic agents (0.15 mg/kg medetomidine hydrochloride, 2.0 mg/kg midazolam, and 2.5 mg/kg butorphanol tartrate in 0.9% NaCl). To prevent uterine muscle contraction during manipulation, 50–80 μl of 0.1 mg/ml ritodrine hydrochloride in 0.9% NaCl was evenly injected into the abdominal cavity. Next, 2–3 μl of plasmid DNA solution in HBS (20 mM Hepes, 150 mM NaCl, pH 7.4) (2–4 mg/ml with 0.01% Fast Green) was injected into the lateral ventricle of the embryo using a mouth-controlled micropipette, and electrical pulses were delivered using a NEPA21 electroporator (voltage: 30–33 V; pulse length: 30 ms; pulse intervals: 970 ms; repeated pulses: 5) with a tweezer-type electrode (CUY650P5; NepaGene, Ichikawa, Japan). After surgery, the mice were warmed on a plate maintained at 38.5˚C with careful monitoring until they woke from the anesthesia. Then, the mice receiving surgery were returned to their original cage and housed together with nesting materials to prevent a decrease of body temperature. Two days before of the expected date, each mouse was transferred to a new cage with part of the nest and nesting materials and housed individually.

### Immunostaining of mouse brain sections

Mice under deep isoflurane anesthesia were fixed by transcardial perfusion with 4% paraformaldehyde (PFA) in 0.1 M PB (0.1 M $NaH_2PO_4$ and 0.1 M $Na_2HPO_4$, pH 7.4). The brains were removed from the skull and post-fixed in the same solution at 4˚C for 4 h with slow rotation. After fixing, the brains were rinsed three times (for 30–60 min each) with 0.1 M PB, immersed in 0.1 M PB containing 0.05% ProClin300 (Sigma-Aldrich, St. Louis, MO) as a preservative, stored at 4˚C, and sectioned for immunostaining within 2 months.

For the preparation of microslice sections, the brains were sectioned at 50–70 μm thickness using a Vibratome (VT1200S; Leica Microsystems, Wetzlar, Germany). Brain slices were collected in 0.1 M PB containing 0.05% ProClin300 and stored at 4˚C.

For immunostaining, brain slices were permeabilized and blocked with 0.1% Triton-X-100 in PBS (8.1 mM $Na_2HPO_4 \cdot 12H_2O$, 1.47 mM $KH_2PO_4$, 137 mM NaCl, 2.7 mM KCl) (PBST) containing 10% fetal calf serum for 30 min at RT with gentle shaking. Next, slices were incubated overnight at 4°C or for 2–3 h at RT with primary antibodies diluted in PBST. Following three washes in PBST (for 10 min each), slices were incubated for 2 h at RT with appropriate secondary antibodies conjugated with Alexa Fluor-488, -555, or -647 (Thermo Fisher Scientific, Waltham, MA, USA) and diluted in PBST.

Brain sections were observed with a Keyence BZ-8000 microscope (Keyence, Osaka, Japan) and an AxioImager Z1 microscope (Carl Zeiss, Oberkochen, Germany) equipped with a CSU10 disc confocal system (Yokogawa, Tokyo, Japan) and OrcaII CCD camera (Hamamatsu Photonics, Hamamatsu, Japan).

Acquired images were processed using ImageJ software (NIH, Bethesda, MD, USA) to appropriately adjust the brightness and contrast. ImageJ software was also used to determine the number of cells.

## Antibodies

For immunostaining, the following antibodies were used: anti-EBNA1 mouse mAb (1:300, NB100-66642), anti-Nestin mouse mAb (1:800, NBP1-92717), and anti-Olig2 rabbit pAb (1:1000, NBP1-28667) (Novus Biologicals, Littleton, CO, USA); anti-GFAP mouse mAb (1:1500, GA5) and anti-Doublecortin rabbit pAb (1:1000, 4604) (Cell Signaling Technologies, Danvers, MA, USA); and anti-MASH1 mouse mAb (1:250, 24B72D11; Thermo Fisher Scientific). For western blotting, the following antibodies were used: anti V5 mouse mAb (1:5000, R960-25; Thermo Fisher Scientific) and anti-GFP mouse mAb (1:1000, sc-9996; Santa Cruz Biotechnology, Santa Cruz, CA, USA).

## Plasmids

For the pCAG-GFP plasmid, DNA fragments encoding the CAG promoter and EGFP (GFP expression cassette) were ligated and cloned into a single plasmid. For pCAG-GFP-EB-oriP, a GFP expression cassette, CMV promoter-driven EBNA1 expression cassette, and oriP [51] were ligated and cloned into a single plasmid. For pCAG-GFP-EB and pCAG-GFP-oriP, either oriP or the EBNA1 expression cassette was removed from pCAG-GFP-EB-oriP, respectively. For pCAG-RFP-EB-oriP, the EGFP coding fragment was replaced with the TurboRFP coding fragment.

For Acceptor shown in Fig 6, loxP-STOP-loxP cassette was inserted between the CAG promoter and the EGFP coding region of pCAG-GFP-EB-oriP, and 5xUAS and the subsequent mCherry coding fragment were ligated and inserted. Sequence of loxP: 5'-TATAACTTCGT ATAGTATACATTATACGAAGTTATAG-3'. Sequence of 5xUAS: 5'-CGGAGTACTGTCCTC CGAGCGGAGTACTGTCCTCCGAGCGGAGTACTGTCCTCCGAGCGGAGTACTGTCCTCCGA GCGGAGTACTGTCCTCCGG-3'. Simian virus 40 (SV40) polyadenylation signal sequence (5'-TTG TTTATTGCAGCTTATAATGGTTACAAATAAAGCAATAGCATCACAAATTTCAC AAATAAAGCATTTTTTTCACTGCATTCTAGTTGTGGTTTGTCCAAACTCATCAATGTAT CTTA-3') was used as a loxP-flanked STOP sequence. For Donor, CAG promoter-driven EBNA1, oriP, mouse GFAP promoter-driven GAL4VP16, and CMV promoter-driven synthetic Cre recombinase [52] were ligated and cloned into a single plasmid. Mouse GFAP promoter was cloned by PCR from genomic DNA of C57BL6J mouse using the forward and reverse primers 5'-GTCTGTAAGCTGAAGACCTGGCAGTG-3' and 5'-CATGGTGCCCTG CCTCTGCTGGCTCCTGGGAT-3', respectively. GAL4VP16 is a fusion protein of the DNA-binding domain of yeast GAL4 (1–147 aa) and the activation domain of viral VP16 (411–490

aa). V5-tagged mouse TBCD expression plasmid was provided by Yokohama City University, Yokohama, Japan.

## Short hairpin RNA (shRNA)

An shRNA plasmid targeting mouse *Tbcd* was constructed using DNA oligos designed with a 21-mer sense sequence, 9-nucleotide loop (ttcaagaga), and 21-mer antisense sequence. Sequence for shTBCD#1: 5′-GGTAACATATCTAACTGTTTCttcaagagaGAAACAGTTAG ATATGTTACC-3′. Sequence for shTBCD#2: 5′-GTCTTGGAGGAGAGCTTATGAttcaaga gaTCATAAGCTCTCCTCCAAGAC-3′. For non-silencing control shRNA, a 19-mer sense sequence, 9-nucleotide loop, and 19-mer antisense sequence were used. Sequence for the control shRNA: 5′-CAGTCGCGTTTGCGACTGGttcaagagaCCAGTCGCAAACGCGACTG-3′.

## Cell culture, transfection, and western blotting

HEK293T (293T) cells were maintained in Dulbecco's modified Eagle's medium supplemented with 10% fetal bovine serum (FBS), under 5% $CO_2$ at 37˚C. For transfection, 293T cells were seeded into a 12-well plate at a density of $2 \times 10^5$ cells/well. Plasmid transfections were performed using polyethyleneimine (PEI). After 24 h, cells were harvested with Laemmli sample buffer supplemented with 0.1 M dithiothreitol (DTT). Harvested samples were sonicated and heated at 70˚C for 20 min, and then separated by SDS-PAGE and analyzed by western blotting, in accordance with standard protocols.

## Supporting information

**S1 Fig. Persistent transgene expression in all known progeny of NPCs by IUE of the EB-oriP plasmid in adult mice.** (A) GFP fluorescence images of the brains transfected at E15 with pCAG-GFP and pCAG-GFP-EB-oriP and fixed at P56. GFP fluorescent images of coronal sections (upper). Magnified images of the boxed regions in upper panels (lower right). Merged images of GFP fluorescence and bright-field image of transfected brains (lower left) used for sections shown in the upper and lower right. Scale bars: 5 mm. Scale bars: 1 mm (upper), 5 mm (lower left), 200 μm (lower right). (B and C) Sagittal sections of the brains transfected with pCAG-GFP (B) and pCAG-GFP-EB-oriP (C) and fixed at P56 were immunostained for DCX. Merged images of GFP fluorescence and bright-field image of the brains used for sections are shown in top left panels. The boxed regions in RMS (left) and OB (right) were magnified and are shown in bottom panels. Scale bars: 5 mm in top left panel, 500 μm in top right panel, 200 μm in bottom panels. (D) Representative magnified images of GFP-labeled neurons in superficial region of the brains transfected with pCAG-GFP and pCAG-GFP-EB-oriP, fixed at P56, and immunostained for EBNA1. Scale bars: 100 μm. (E) Representative GFP-labeled cells appeared to be an astrocyte (i), a migrating neuroblast in RMS (ii), neurons in OB (iii), and an ependymal cell and astrocytes in V-SVZ (iv) in the section of pCAG-GFP-EB-oriP-transfected brain fixed at P56 and immunostained for EBNA1. Scale bars: 25 μm in (i), (ii), (iv), 50 μm in (iii).
(PDF)

**S2 Fig. Sequences encoding both EBNA1 and oriP in the plasmid for IUE are required for transgene expression in postnatal progeny of NPCs.** (A) Schematic illustration of plasmids encoding either EBNA1 (pCAG-GFP-EB) or oriP (pCAG-GFP-oriP), in addition to GFP. *In utero* electroporation into the lateral ventricle was performed at E15 and the brains were analyzed at P21 and P56. (B and C) (top) GFP fluorescence images of coronal sections from transfected brains at P21 (B) and P56 (C). (middle) Magnified images of the boxed regions in top

panels. (bottom) Merged images of GFP fluorescence and bright-field image of transfected brains used for the sections shown above. Scale bars: 1 mm in top panels, 200 μm in middle panels, 5 mm in bottom panels. (D) Representative magnified images of GFP-labeled neurons in superficial region of the brains transfected with pCAG-GFP-EB (left) and pCAG-GFP-oriP (right) at P56 and immunostained for EBNA1. Scale bars: 100 μm. (E and F) Sagittal sections of the brains at P21 (E) and P56 (F) transfected with pCAG-GFP-EB (left, respectively) and pCAG-GFP-oriP (right, respectively) were immunostained for DCX. Merged images of GFP fluorescence and bright-field image of the brains used for sections are shown in upper left panels. The boxed regions in RMS (left) and OB (right) were magnified and are shown in lower panels. Scale bars: 5 mm in upper left panel, 500 μm in upper right panel, 200 μm in lower panels.
(PDF)

**S3 Fig. Transgene expression in not only neurons but also other postnatal progeny of NPCs by IUE of the EB-oriP plasmid in young adult C57BL6J mice.** (A) Merged images of GFP fluorescence and bright-field image of the brains transfected with pCAG-GFP (left) and pCAG-GFP-EB-oriP (right). *In utero* electroporation into the lateral ventricle of C57BL6J mice was performed at E15 and observed at P21. Scale bars: 5 mm. (B) GFP fluorescence images of coronal sections from transfected brains shown in (A). Scale bars: 1 mm. (C) Magnified images of the boxed regions in (B). Arrows indicate GFP-labeled cells that morphologically appeared to be astrocytes. Scale bars: 200 μm. (D and E) Sagittal sections of the brains transfected with pCAG-GFP (E) and pCAG-GFP-EB-oriP (F) were immunostained for DCX. Merged images of GFP fluorescence and bright-field image of the brains used for sections are shown in upper left panels. The boxed regions in RMS (left) and OB (left) were magnified and are shown in lower panels. Scale bars: 5 mm in upper left panel, 500 μm in upper right panel, 200 μm in lower panels.
(PDF)

**S4 Fig. Human β-interferon S/MAR sequence in the plasmid for IUE has little effect on persistent and inheritable transgene expression in postnatal progeny of NPCs in ICR mice.** (A) Schematic illustration of the plasmid encoding human β-interferon S/MAR sequence, in addition to GFP. (B) A GFP fluorescence image of the brain transfected at E15 with pCAG-GFP-S/MAR and fixed at P21. A GFP fluorescent image of the coronal section (upper). A magnified image of the boxed region in upper panel (lower right). A GFP fluorescence image of the transfected brain (lower left) used for sections shown in the upper and lower right. Scale bars: 5 mm. Scale bars: 1 mm (upper), 5 mm (lower left), 200 μm (lower right). (C) A sagittal section of the brain at P21 transfected with pCAG-GFP-S/MAR was immunostained for DCX. A GFP fluorescence image of the brain used for sections are shown in the upper left panel. The boxed region in RMS (left) and OB (right) were magnified and are shown in lower panels. Scale bars: 5 mm in upper left panel, 500 μm in upper right panel, 200 μm in lower panels.
(PDF)

**S1 Raw images.**
(PDF)

# Acknowledgments

The author thanks Drs. Atsushi Suzuki, Yukio Sasaki, and Yutaka Kawakami for comments on the manuscript; Dr. Allen Yi-Lun Tsai for helping prepare the manuscript; Mr. Yuki

Kawaguchi, Mr. Junpei Matsubayashi, and Ms. Eriko Saito for discussions; Drs. Noriko Miyake and Naomichi Matsumoto for providing the mouse TBCD expression plasmid; Dr. Yoshihiro Miwa for providing the plasmid containing EBNA1 and oriP; and Dr. Atsushi Suzuki for providing the experimental facilities. The author also thanks Edanz (https://jp.edanz.com/ac) for editing a draft of this manuscript.

## Author Contributions

**Conceptualization:** Tomoko Satake.

**Data curation:** Tomoko Satake.

**Funding acquisition:** Tomoko Satake.

**Investigation:** Tomoko Satake.

**Writing – original draft:** Tomoko Satake.

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
