## [Decision Letter · Decision Letter 0]

30 Jul 2021

PONE-D-21-19501

Epstein–Barr virus-based plasmid enables inheritable transgene expression in mouse cerebral cortex

PLOS ONE

Dear Dr. Satake,

Thank you for submitting your manuscript to PLOS ONE. After careful consideration, we feel that it has merit but does not fully meet PLOS ONE’s publication criteria as it currently stands. Therefore, we invite you to submit a revised version of the manuscript that addresses the points raised during the review process.

Both reviewers and I, we very much like the study. However, the n-number in the most important experiments is too low. Both reviewers open different options to solve this major point by either increasing the n-numer to four animals for statistical testing of the most important data sets (see the comments of both reviewers). Other data that are less relevant regarding the properties of the plasmids, meaning the data that are more descriptive, might be presented as single animal data (e.g. by showing representative images or countings for both animals). The focus should be put on the properties of the plasmids, as this is the main topic of the paper.

Furthermore, please add more details about the post-surgery treatment (medication, analgesic treatment) of the experimental animals. 

We look forward to receiving your revised manuscript.

Kind regards,

Robert Blum

Academic Editor

PLOS ONE

Reviewers' comments:

Reviewer's Responses to Questions

**Comments to the Author**

1. Is the manuscript technically sound, and do the data support the conclusions?

Reviewer #1: Partly

Reviewer #2: Yes

2. Has the statistical analysis been performed appropriately and rigorously? 

Reviewer #1: No

Reviewer #2: N/A

3. Have the authors made all data underlying the findings in their manuscript fully available?

Reviewer #1: Yes

Reviewer #2: Yes

4. Is the manuscript presented in an intelligible fashion and written in standard English?

Reviewer #1: Yes

Reviewer #2: Yes

5. Review Comments to the Author

Reviewer #1: This is an interesting manuscript describing the use of Epstein-Barr virus-based plasmids to study neocortical development in mice. By directly comparing the populations of transfected cells in mice electroporated with either pCAG-GFP or pCAG-GFP-EB-oriP plasmids, the authors clearly show that the second plasmid labels a larger population of neurons, astrocytes and postnatal SVZ progenitors, which could be likely explained by the increased stability/inheritability of the EB-oriP transgene. Overall, the manuscript is presented in an intelligible fashion, figures are of good quality and results presented are of potential interest to the field, especially from a technical perspective. However, there are some important points that I would like to see addressed before the possible publication of the manuscript:

1) Quantifications shown in figures 1, 4, 5 and 7 were performed only in 2 animals. This is very low, considering the high variability of experiments using in utero electroporations (IUE). Moreover, this sample size does not allow statistical comparisons.

I understand the amoun of work involved in these experiments and don't expect that the authors increase the sample size for all those 4 different quantifications. However, the authors must definitely increase the sample sizes if they want to perform statistical comparisons. Otherwise, they must describe the data per animal.

In my opinion, an interesting approach could be:

Figure 1 and 4: increase samples sizes and perform statistical tests, since data described in these figures are important to validadte the benefit of using the EB-oriP plasmid.

Figure 5: show results per animals. As I will detail below, the data described here is interesting, but can have different interpretations.

Figure 7: increase sample size in one time-point (P7 or P14) and perform statistical test.

2) Electroporation of CreERT2 plasmids in reporter mouse lines has been largely used to stably label neocortical progenitors and their progeny through Cre-lox recombination. This can be combined with cell-type specific promoters and shRNAs, as described in the current manuscript for the EB-oriP. The authors should discuss the pros and contras of this latter method in comparison with Cre-lox based strategies.

3) Results described in Figure 7 are interesting, but highly preliminar. To support the interpretation that neurons transfected with shRNA against TBCD have a reduced survival, authors must provide more evidence, such as similar numbers of transfected neurons at earlier time-points (P0), increased proportions of dying neurons in the first postnatal week, etc.

4) The discussion about progenitor heterogeneity needs to be tuned down. An alternative explanation for the observations described in Figure 5 could be that only a small fraction of NPCs transfected at E13 would remains as NPCs at E15 and generate upper layer neurons and astrocytes, whereas the vast majority of E13 progenitors would generate deep layer neurons and be depleted. Therefore, the low frequency of co-transfceted cells in the "separate electroporation" experiment could be explained by the low number of electroporated cells analysed. It is important to note here that every 1000 cells counted at P21 could indicate the the progeny of less than 100 NPCs electroporated at E13/15. Moreover, as pointed out above, these quantifications were made only in 2 animals.

Reviewer #2: The present manuscript reports the use of an Epstein-Barr virus-based plasmid to label cells and their progeny over several divisions, from embryonic neurogenesis to, at least, young adulthood (P56). This works both with constitutive and cell-specific promoters, and allows the manipulation of the gene expression. Furthermore, the author shows that progenitor cells have a different susceptibility to electroporation at different embryonic stage. The system is interesting because its components don’t seem to interfere with cell behavior, and the plasmid remains episomal, avoiding the risks connected to genome integration, as the case for transposons or virus-mediated fate mapping.

The experiments are well designed, well performed and analyzed; the text is well written and clear.

There are some major points and a minor point to be discussed and addressed:

_ for most experiments, the number of biological replicates is 2 (Figures 1, 4, 5, 7) and this appears a low number. As many timepoints have been analyzed, it would be important to add at least one more biological replicate to the most important experiments, e.g., Figure 1F, Figure 4C.

_ along this line, a section describing the statistical analysis (e.g., how many sections per animal?) would be helpful.

_ In the use of cell-specific promoters (mGFAP, Figure 6), some GFP+ cells (electroporated) are also RFP (supposedly, GFAP+): can the author quantify the proportion of RFP+/GFP+ over time (P5, P14 and P21)? This because it’s known that in the mouse cortical gray matter GFAP expression declines with mice becoming adult. This would indeed further support the use of this system as tool to investigate physiological properties of brain cells.

_ in the TBCD knock-down experiment (Figure 7), a progressive reduction of GFP expression is observed. Can the author stain for EBNA1 (like in figure S1E) to ascertain whether the loss of signal is not due to a reduction in GFP expression, but a loss of cells? If that’s the case, can the author analyze cell death between P1 and P7 (e.g., P3 or P4)? This would reinforce the hypothesis that TBCD knock-down causes the loss of cells.

_ this system seems particularly suitable for clonal analysis: has the author tried to electroporate ng (instead of microgram) of plasmidic DNA, and assess whether clonality can be observed (e.g., 1 or very few progenitor cells giving rise to intermediate progenitors, neurons, and astrocytes)? This could be a potential application worth discussing.

Minor point:

_ can the author explain in the figure legends when pseudo-color is used? For instance, in Figure 2E, the big panel shows green (GFP) and violet (Dcx): however, in the RMS violet and white color are shown. Likewise, in Figure 2F, Figure 4E and 4F.

6. PLOS authors have the option to publish the peer review history of their article (what does this mean?). If published, this will include your full peer review and any attached files.

Reviewer #1: **Yes: **Marcos R. Costa

Reviewer #2: No

---

## [Author Response · Author response to Decision Letter 0]

20 Aug 2021

I am grateful to the reviewers and editor for critical reading of the manuscript and for providing insightful comments. The manuscript has been improved by the helpful suggestions. Most of the suggestions provided have been incorporated into the revised manuscript.

The point-by-point responses to the issues raised are provided as a separate file.

---

## [Decision Letter · Decision Letter 1]

17 Sep 2021

Epstein–Barr virus-based plasmid enables inheritable transgene expression in mouse cerebral cortex

PONE-D-21-19501R1

Dear Dr. Satake,

We’re pleased to inform you that your manuscript has been judged scientifically suitable for publication and will be formally accepted for publication once it meets all outstanding technical requirements.

Kind regards,

Robert Blum

Academic Editor

PLOS ONE

Additional Editor Comments (optional):

Dear Authors,

Reviewer 2 came up with a small additional aspect which does not affect the content or scientific impact of your study. He suggests to re-organize the statistics (see his comment). It is your decision whether you would like to do this or not. I think that there are good arguments to keep the statistics in the text and there are good arguments to organize them in the material and method section. Anyhow, thank you very much for your study.

Best

Robert Blum (academic editor)

Reviewers' comments:

Reviewer's Responses to Questions

**Comments to the Author**

1. If the authors have adequately addressed your comments raised in a previous round of review and you feel that this manuscript is now acceptable for publication, you may indicate that here to bypass the “Comments to the Author” section, enter your conflict of interest statement in the “Confidential to Editor” section, and submit your "Accept" recommendation.

Reviewer #1: All comments have been addressed

Reviewer #2: All comments have been addressed

2. Is the manuscript technically sound, and do the data support the conclusions?

Reviewer #1: Yes

Reviewer #2: Yes

3. Has the statistical analysis been performed appropriately and rigorously? 

Reviewer #1: Yes

Reviewer #2: Yes

4. Have the authors made all data underlying the findings in their manuscript fully available?

Reviewer #1: Yes

Reviewer #2: Yes

5. Is the manuscript presented in an intelligible fashion and written in standard English?

Reviewer #1: Yes

Reviewer #2: Yes

6. Review Comments to the Author

Reviewer #1: The authors have adequately addressed my main concerns. I would just suggest that they add a normalized quantification of the cell populations (Layer II-III, Layer IV, Layer V and astrocytes) labeled with GFP, RFP or both flurescent markers in Figure 5G. This will facilitate the interpretation of the results described there.

Reviewer #2: The author addressed the main point of the reivision, namely the low number of animals analysed. About the other points, the authors added comments in the discussion when needed.

Even if the author added the number of sections quantified in the figure legend, I would encourage the author to add a section "Quantification and Statistical Analysis" in the material and methods, as it would be easier for the reader find this information in a specific section and not throughout the manuscript.

7. PLOS authors have the option to publish the peer review history of their article (what does this mean?). If published, this will include your full peer review and any attached files.

Reviewer #1: **Yes: **Marcos R. Costa

Reviewer #2: No

---

## [Editor Report · Acceptance letter]

22 Sep 2021

PONE-D-21-19501R1 

Epstein–Barr virus-based plasmid enables inheritable transgene expression in mouse cerebral cortex 

Dear Dr. Satake:

I'm pleased to inform you that your manuscript has been deemed suitable for publication in PLOS ONE. Congratulations! Your manuscript is now with our production department. 

Kind regards, 

on behalf of

PD Dr. Robert Blum 

Academic Editor

PLOS ONE